# Projected Distribution and Dispersal Patterns of Potential Distribution *Fasciola hepatica* and Its Key Intermediate Host *Radix* spp. in Qinghai-Tibet Plateau, China, Under Plateau Climatic Conditions

**DOI:** 10.3390/pathogens14070647

**Published:** 2025-06-30

**Authors:** Luyao Xu, Yunhai Guo, Zengkui Li, Mingjia Guo, Ming Kang, Daoxin Liu, Limin Yang, Zhongqiu Li, Panpan Wang, Wenhui Luo, Ying Li

**Affiliations:** 1College of Agriculture and Animal Husbandry, Qinghai University, Xining 810016, China; xuluyao0214@163.com (L.X.); lizengkui0502@163.com (Z.L.); 2013990003@qhu.edu.cn (M.K.); liudaoxin_dy2007@163.com (D.L.); 18130690165@163.com (P.W.); 18024842902@163.com (W.L.); 2Qinghai Provincial Key Laboratory of Pathogen Diagnosis for Animal Diseases and Green Technical Research for Prevention and Control, Xining 810016, China; 3National Institute of Parasitic Diseases at Chinese Center for Disease Control and Prevention, Chinese Center for Tropical Diseases Research, Shanghai 200025, China; guoyh@nipd.chinacdc.cn (Y.G.); yanglm@nipd.chinacdc.cn (L.Y.); lizq@nipd.chinacdc.cn (Z.L.); 4Animal Disease Prevention and Control Center, Jinan 250100, China; 15098615210@163.com

**Keywords:** *Fasciola hepatica*, *Radix*, Maxent, Qinghai-Tibet Plateau, animal husbandry

## Abstract

(1) Background: As a prominent zoonotic parasitic disease, fascioliasis threatens the sustainable development of animal husbandry and public health. Current research focuses mainly on individual species (parasite or intermediate host), neglecting systematic evaluation of the transmission chain and exposure risks to animal husbandry. Thus, comprehensive studies are urgently needed, especially in the ecologically fragile alpine region of the Qinghai-Tibet Plateau; (2) Methods: Distribution data of *Radix* spp. and *Fasciola hepatica* in the Qinghai-Tibet Plateau and adjacent areas were gathered to establish a potential distribution model, which was overlaid on a map of livestock farming in the region; (3) Results: The key environmental factors influencing *Radix* spp. distribution were temperature seasonality (21.4%), elevation (16.4%), and mean temperature of the driest quarter (14.7%). For *F. hepatica*, the main factors were elevation (41.3%), human footprint index (30.5%), and Precipitation of the driest month (12.1%), with all AUC values exceeding 0.9. Both species exhibited extensive suitable habitats in Qinghai and Tibet, with higher *F. hepatica* transmission risk in Qinghai than Tibet; (4) Conclusions: The significant transmission risk and its impacts on the livestock industry in the Qinghai-Tibet Plateau highlight the need for proactive prevention and control measures. This study provides a scientific foundation for targeted alpine diseases control, establishes an interdisciplinary risk assessment framework, fills gaps in high-altitude eco-epidemiology, and offers insights for ecological conservation of the plateau.

## 1. Introduction

The Lymnaeidae family (Mollusca, Gastropoda, Pulmonata, Basommatophora), commonly known as pond snail [1], has a global distribution. In China, three genera are identified: *Lymnaea*, *Radix*, and *Galba*. These snails serve as intermediate hosts for numerous humans and livestock parasites, causing parasitic diseases that threaten public health and incur significant economic losses [2]. Fascioliasis, a zoonotic disease caused by species of the *Fasciola* genus, primarily infects herbivores like cattle and sheep, occasionally damaging human hepatobiliary systems. *Fasciola* species belongs to Platyhelmintha, Trematoda, Digenea, and Fasciolidae, with *Fasciola hepatica* and *Fasciola gigantica* being most detrimental to human and animal health [3]. *F. hepatica* infects approximately 20 million people worldwide, posing major public health challenges [4]. In China, fascioliasis is endemic in pastoral regions such as Inner Mongolia, Qinghai, and Ningxia [5]. Lymnaeidae snails are crucial for completing the *F. hepatica* life cycle, with at least 20 species acting as potential vectors [6]. *Radix* spp. are the most abundant and widely distributed in China [7]. For example, *Radix cucunorica* was identified as the primary intermediate host of *F. hepatica* in the Gannan pastoral region [8], while *Radix* spp. dominate in Qinghai and Tibet [9,10]. These snails exhibit remarkable adaptability, thriving in diverse habitats from thermal spring habitats to aquatic systems at altitudes over 3000 m [11]. Suitable snail habitats are essential for *Fasciola* development, making Lymnaeidae distribution patterns critical for understanding fascioliasis spread.

The niche model integrates species distribution data with environmental datasets, employing algorithms to determine ecological requirements. It projects findings across temporal and spatial scales to predict actual and potential species distributions [12]. This approach is widely utilized to investigate interplay between infectious diseases spread and environmental variables, enabling assessment of transmission dynamics and risks [13]. Recently, it has been adapted to forecast transmission risks posed by key hosts or vectors, categorizing parasitic diseases based on potential distribution likelihood [14]. For example, niche models have evaluated climate change impacts on *Oncomelania hupensis* (a *Schistosoma mansoni* vector in Hubei Province), predicting parasite transmission risk [15,16]. Similarly, they identified *S. mansoni* risk areas using the intermediate host *Biomphalaria straminea* distribution data [17]. International researchers have also used intermediate hosts distributions to predict epidemic risks for *Plasmodium knowlesi* and canine leishmaniosis [18,19,20].

Additionally, niche models aid in assessing vector habitat adaptability. Certain studies have used these models to evaluate suitable habitats for *Pomacea canaliculata* (a key intermediate host for *Angiostrongylus cantonensis*), clarifying correlation between vector distribution and environmental factors [21,22]. Building on this, our research pioneers the use of ecological niche models (ENMs) to assess transmission risks of *F. hepatica* and its intermediate host *Radix* spp., evaluating their potential geographic ranges and infection risk. Among modeling instruments, the MaxEnt model is favored for its exceptional precision, user-friendliness, and concise output [23,24]. The receiver operating characteristic (ROC) curve’s area under the curve (AUC) possesses significant benefits, providing threshold-independent evaluation, which has driven its extensive adoption in model assessment [25]. Thus, with a strong correlation between vectors and environmental conditions, MaxEnt enables invaluable predictions of species distribution under changing conditions, guiding targeted control strategies.

High-altitude regions are highly sensitive to global climate change. The Qinghai-Tibet Plateau, known as the world’s highest plateau, possesses a unique natural environment and complex topography. These characteristics foster rich biodiversity and vector species proliferation [26]. Climate factors significantly influence vector spatio-temporal pattern and pathogens survival [27]. The plateau is projected to experience continued warming and increased moisture [28].

Drawing distribution data of *Radix* spp. and *F. hepatica* in the Qinghai-Tibet Plateau and adjacent areas, we correlated environmental factors with their prevalence. Integrating local animal husbandry practices, the model identified high-risk zones. These findings provide a scientific basis for parasitic diseases control, laying groundwork to curb diseases spread on the plateau.

## 2. Materials and Methods

### 2.1. Species Distribution Data Collection

Distribution data for *Radix* spp. and *F. hepatica* in the Qinghai-Tibet Plateau and adjacent regions were collected by referencing the Global Biodiversity Information Facility (GBIF) database (http://data.gbif.org/welcome.html; accessed on 16 August 2024), relevant literature, and publications. Specifically, GBIF was searched for “*Radix* spp.” and “*Fasciola hepatica*” with filters: “in China”, “1980 to present”, “including coordinates”, and “present”. Record details were checked to verify classification consistency. Additionally, CNKI was searched for “Lymnaeidae” and “*F. hepatica*”, with original data selected based on morphological and molecular identification of “*Radix* spp.” and “*F. hepatica*”. Where latitude and longitude were not provided, Google Earth supplemented the information. This yielded 243 distribution records for *Radix* spp. (refer to Appendix A) and 96 for *F. hepatica* were compiled (refer to Appendix A). The ENMtools was applied to eliminate duplicate and over-fitting data points, mitigating sampling bias. Refined datasets 226 of *Radix* spp. and 79 of *F. hepatica* records were utilized for modeling. The MaxEnt is particularly appropriate for small sample datasets (where n ≥ 15) and performs optimally with approximately around 120 samples [29,30], ensuring robust results from this study’s dataset.

### 2.2. Relevant Environment Variables Collection and Filtering

Following an extensive literature review, a detailed analysis of the biological attributes of *Radix* spp. was conducted. Table 1 below describes the definitions and sources of these datasets. Utilizing coordinate data on the distribution of *Radix* spp. and *F. hepatica* across the Qinghai-Tibet Plateau and adjacent regions, corresponding environmental variable data were meticulously extracted. Notably, the complexity of the Maxent model significantly impacts its transferability; higher complexity tends to negatively impact model performance [31,32]. Complicated models often overfit species distribution data within the modeling domain, compromising predictive accuracy, leading to substantial deviations from actual conditions and yielding unreliable results [33].

To address this, the environmental variables were extracted using a mask in ArcGIS 10.8 software, referencing China’s standard administrative division map (Review No. GS (2024) 0650). Subsequently, the SDMToolbox was engaged to perform a correlation analysis on variables, and variables with a Pearson correlation coefficient < 0.9 was selected for model inclusion.

### 2.3. Analytical Tools and Methods

In this study, ENM was used to map distributional origins. The ENM was developed via the maximum entropy model (MaxEnt) implemented through the software available at http://www.cs.princeton.edu (accessed on 9 March 2023). Comparative evaluations of diverse algorithmic models suggest that MaxEnt, which incorporates machine learning concepts and the maximum entropy algorithm, provides more accurate forecasts. This method is deemed the most effective for species distribution modeling [29,30]. For model construction, 75% of the data were allocated to training, with the remaining 25% reserved for validation.

### 2.4. Model Parameter Tuning Using the ENMeval Package

The complexity of the MaxEnt model significantly influences its transferability. Highly complex models overfit species distribution data during training, reducing predictive accuracy when transferred and leading to simulations that deviate from reality and compromise reliability. MaxEnt model complexity depends on its Regularization Multiplier (RM) and Feature Combination (FC), which includes five features: linear-L, quadratic-Q, hinge-H, product-P, and threshold-T. Feature combination selection relates to the number of species distribution points. The ENMeval package evaluates model complexity by analyzing MaxEnt’s corrected Akaike information criterion correction (AICc) across diverse parameter settings. AIC measures overfitting propensity, favoring models with the lowest values [34]. Parameter tuning via ENMeval shows that lower-complexity models exhibit better spatial transferability and more accurate predictions of species potential distributions [33].

### 2.5. Model Accuracy Verification

The Area Under Curve (AUC), a widely utilized research metric, evaluates model accuracy [35]. Ranging from 0 to 1, an AUC of 0.5 to 0.7 indicates moderate accuracy and potential reliability issues. A score of 0.7 to 0.9 suggests higher accuracy but requires careful examination, while 0.9 to 1 indicates high accuracy and considerable dependability. Conversely, an AUC of 0.5 indicates random prediction [36].

### 2.6. Analyze the Impact of Environmental Variables

The pivotal environmental variables and their distinct impacts on *Radix* spp. and *F. hepatica* occurrences were meticulously assessed utilizing the jackknife method. Following this analysis, the potential habitats for *Radix* spp. and *F. hepatica* were identified through an examination of the response curve derived from MaxEnt.

### 2.7. Possibility of Predicting the Distribution of Radix spp. and Fasciola hepatica

In ArcGIS 10.8, the MaxEnt output results were utilized to visualize. Using Jenks’ natural breaks classification, we reclassified risk values into four levels: high-risk, medium-risk, low-risk, and non-risk areas. An ascending gradient indicates an increased likelihood of *Radix* spp. and *F. hepatica* distribution. This method optimally groups values by iteratively comparing within-group and between-group mean differences. In transmission risk assessment, it objectively characterizes probability distributions while minimizing human-induced errors [37], making it widely used in vector-borne diseases risk assessment [38]. Additionally, the ArcGIS 10.8 software’s toolset was utilized to calculate the area of each risk area.

### 2.8. Livestock Impact Assessment

The previously mentioned results were superimposed upon the distribution map of the Qinghai-Tibet Plateau, as illustrated in Figure 1 [39]. In light of these derived insights, an analysis of the implications for animal husbandry within the region was conducted.

## 3. Results

### 3.1. Distribution of Radix spp. and Fasciola hepatica in Qinghai-Tinbet Plateau and Surrounding Areas

Yang indicates that in limpets studies, the high plasticity of the shell morphology of this group and the long-standing controversy have made species identification both crucial and challenging. The study concluded that there are species of the *Radix* genus in the Qinghai region, such as *R. lagotis* and *R. cucunorica*; in the Tibet region, the records include *R. auricularia*, *R. ovata*, *R. lagotis*, and *R. cucunorica* of the *Radix* genus, and *Galba truncatula* of the *Galba* genus [7]. Following this, this study collected *Radix* distribution data based on the article’s classification criteria. After meticulous sorting, we obtained distribution records for *Radix* spp. (see Appendix A) and *F. hepatica* (see Appendix A). Removing duplicate and overfitted data points, the remaining dataset was employed to delineate their comprehensive distribution across the Qinghai-Tibet Plateau and adjacent regions, as depicted in Figure 2. And it also summarized its distribution situation in the more widely-populated regions of Qinghai and Tibet (Table 2).

The findings indicated that *Radix* spp. snails mainly concentrate in the southwest and northeast of the Qinghai-Tibetan Plateau, particularly in the Xizang Autonomous Region and Qinghai Province, with a notably higher density in the former. In contrast, *F. hepatica* is predominantly encountered in the east of the Qinghai-Tibet Plateau, with the widest distribution in the northeast of Qinghai Province.

### 3.2. Environmental Factor Screening of Distribution Prediction Model

In ArcGIS 10.8, a suite of environmental factors, namely Bio1–19, Elev, Hii, and Hfp underwent initial processing. Correlation analysis results are shown in Figure 3A,B.

The correlation coefficient strength reflected interrelation among variables. First, variables prone to over-fitting (including Bio8, Bio9, Bio18, and Bio19) were excluded. Furthermore, variables with an absolute value of the correlation coefficient exceeding 0.9 and other variables (shown in dark red and dark blue, respectively, in Figure 3) were excluded. Ultimately, for *Radix* spp., selected variables (Bio1, 2, 3, 4, 5, 7, 12, 14, 15, Elev, and Hfp) construct the potential distribution model (Figure 3A). For *F. hepatica*, the model used Bio1, 2, 3, 6, 12, 14, 15, Elev, and Hfp (Figure 3B).

### 3.3. Model Parameter Tuning Using the ENMeval Package

The ENMeval package was meticulously evaluated, with the outcomes presented in Figure 4. The *Radix* spp. potential distribution model exhibits optimal complexity using parameters RM 0.5 and the FC LQHPT, yielding an AICc of 0 (Figure 4A). For *F. hepatica*, the preferred parameters were RM 1.5 and FC LQH, also giving an AICc value of 0 (Figure 4B). These parameters promote lower model overfitting and reduce prediction errors during modeling.

### 3.4. Evaluation of Distributed Prediction Models

As illustrated in Figure 5, the AUC for the *Radix* spp. distribution model was 0.932 (Figure 5A), while the *F. hepatica* model achieved an AUC of 0.962 (Figure 5B). Both values exceed the random distribution (AUC = 0.5) benchmark and the 0.9 threshold, indicating high prediction accuracy and reliability. Furthermore, the models also effectively outlined a well-defined potential habitat zone in the study area.

### 3.5. Importance Analysis of Environmental Variables

Before obtaining final results, MaxEnt evaluated environment variables’ relative importance and contribution to facilitate variable selection (Figure 6). As shown in Table 3, key factors for *Radix* spp. distribution were temperature seasonality (Bio4, contribution rate 21.4%), elevation (Elev, contribution rate 16.4%), and mean temperature of the driest quarter (Bio9, contribution rate 14.7%). For *F. hepatica*, the most significant variables were elevation (Elev, contributing a proportion of 41.3%), human footprint index (Hfp, 30.5%), and precipitation of the driest month (Bio14, 12.1%) (Table 4). Figure 6 shows the snail model performed best with Elev alone but deteriorated significantly without Hfp (Figure 6A). Similarly, the *F. hepatica* model was optimized with Elev, but accuracy dropped most when Hfp was excluded (Figure 6B), highlighting elevation and human footprint index as critical for both models.

Response curve in Figure 7 indicates *Radix* spp. had the highest occurrence probability at a bio4 value of 375 (Figure 7A) and an elevation of 4727.2 m (Figure 7B), increasing gradually from sea level to this altitude before declining.

The response curve of Bio9 illustrates that the highest occurrence probability for *Radix* spp. at 10 °C, decreasing with rising isothermality values. Conversely, the relationship becomes inverse when temperatures exceed 10 °C (Figure 7C). Survival probability increases within 0 to 2571.4 m elevation and decreases above this threshold (Figure 7D). For the human footprint index, *F. hepatica* presence probability is negligible at index zero, increasing with index elevation (Figure 7E). Furthermore, Bio14 is crucial for *F. hepatica* distribution, with occurrence peaking at −17.5 to 0 mm and then decreasing negatively (Figure 7F). Collectively, these results define the environmental characteristics of suitable habitats for *F. hepatica* and its intermediate host, *Radix* spp.

### 3.6. Prediction of Communication Risk

Ultimately, the models for *Radix* spp. and *F. hepatica* were extrapolated to the Qinghai-Tibet Plateau, reclassifying MaxEnt results into four risk categories. *Radix* spp. risk values were divided into high risk (0.50~1), medium-risk (0.25~0.49), low-risk (0.11~0.24), and non-risk (0~0.10). Similarly, *F. hepatica* risk zones were demarcated into high risk (0.51~1), medium risk (0.26~0.50), low risk (0.11~0.25), and non-risk (0~0.10). Results showed relatively suitable regions in the south, center, and east of the Qinghai-Tibet Plateau (Figure 8), widely covering Qinghai, Tibet, Xinjiang, Yunnan, Gansu, and Sichuan provinces. Tibet had the highest occurrence probability.

Furthermore, integrating the Qinghai-Tibet Plateau river system on the distribution map reveals that *Radix* spp. suitable habitats mainly cluster near riverine systems (see Figure A1). These habitats exhibit high adaptability in the upper reaches of the Yellow River, Yangtze River, and Lancang River in Qinghai Province. Using ArcGis 10.8 to quantify each suitable area, the Qinghai-Tibet Plateau has the following characteristics: high-risk area encompasses 14.1602 × 10^4^ km^2^, the medium-risk area covers 33.5264 × 10^4^ km^2^, the low-risk area extends over 58.0277 × 10^4^ km^2^, and the non-risk area spans 146.3503 × 10^4^ km^2^. These data suggest an elevated transmission risk of *Radix* spp. in the Tibet region.

Figure 9 shows that *F. hepatica* primarily concentrates in the east and north of the Qinghai-Tibet Plateau, with higher suitability in eastern and central Qinghai Province. Overlaying hydrological systems on the distribution map highlights its suitable habitats along riverine corridors (see Figure A2), particularly in Qinghai’s Sanjiangyuan region, the source area of the Yangtze River, the Yellow River, and the Lancang River. Quantified habitat areas on the plateau are as follows: 8.5757 × 10^4^ km^2^ in the highly suitable category, 15.9835 × 10^4^ km^2^ in the medium risk category, 32.4845 × 10^4^ km^2^ in the low-risk area, and 195.021 × 10^4^ km^2^ in the non-risk area. These findings suggest higher *F. hepatica* transmission risk in Qinghai.

### 3.7. Evaluation of the Impact of Radix spp. And Fasciola hepatica on Animal Husbandry in Qinghai-Tibet Plateau

The projected distribution maps of *Radix* spp. and *F. hepatica* were superimposed upon the Qinghai-Tibet Plateau’s livestock distribution map (Figure 10 and Figure 11). Figure 10 depicts grazing areas of Hequ horse and Tibetan goat in Qinghai Province overlapping with zones for *Radix* spp.—the primary intermediate host for *F. hepatica*—indicating elevated transmission risk. Specifically, model projections show that the Tibetan goats in the Yushu Tibetan Autonomous Prefecture and Hequ horses in Huangnan, Hainan Tibetan Autonomous Prefectures, and Haidong region are more susceptible to *Radix* spp. transmission. In Tibet, *Radix* spp. distribution threatens the Xizang *Bos grunnien* in the southern Naqu. The high *Radix* spp. prevalence in Changdu impacts Southern Tibetan sheep husbandry, while their presence in Lasha severely affects Northern Tibetan sheep breeding. These also harm the economies of the Southern Tibetan sheep and Xizang *Bos grunnien* husbandry in Shannan and Ali regions.

Figure 11 similarly indicates Northern Tibetan sheep and Hequ horses overlapping with high-risk zones for *F. hepatica* infestation. In Qinghai province, Hequ horses in the Huangnan, Hainan Tibetan Autonomous Prefecture, and in Haidong City, along with North of Tibet sheep in the Haibei Tibetan Autonomous Prefecture, face elevated F. hepatica transmission risk. In Tibet, the Northern and Southern Tibetan sheep in Lasa and Shannan regions are under significant parasitic threat.

Table 5 indicates that *Radix* spp. pose considerable risks to Hequ horses and Tibetan goats breeding in Qinghai, as well as to Xizang *Bos grunniens* and Southern Tibetan sheep husbanry. Concurrently, *F. hepatica* threatens North Tibetan Tibet sheep, Hequ horse, and Tibetan goats in Qinghai, with Southern Tibetan sheep in Tibet being particularly vulnerable.

### 3.8. Distribution Results of Other Important Trematoda

As carriers of fascioliasis and clonorchiasis, *Fasciola gigantica* and other digenean parasites use Lymnaeidae as key intermediate hosts. Through meticulous data sorting, we have acquired the nationwide distribution data for 44 records on fascioliasis caused by *F. gigantica* (see Appendix A) and 114 records on infections by several species of *Digenea trematodes* (see Appendix A). As depicted in Figure 12, these parasites are mainly distributed in southern and central China, with sporadic records in the Qinghai-Tibet Plateau and adjacent regions. Due to limited local distribution data, this study does not model the plateau, indicating fascioliasis as the predominant trematode disease there. Previous studies reported intermediate-shaped trematodes in Yunnan and Guangxi [40,41] and Qinghai’s Yushu area [42]. However, incomplete research on Qinghai’s intermediate-shaped trematodes mean that the potential impacts cannot be ruled out, limiting this study’s scope.

### 3.9. Distribution Results of Other Important Lymnaeidae

Using the same methodology, 33 records regarding the distribution of the genus *Galba* were systematically compiled (see Appendix A), as shown in Figure 13. The genus *Galba* is predominantly distributed in Guangxi Province, with sparse records in Tibet and near absence in Qinghai. Previous studies suggested that dominant snail species suppress others [9]. In Qinghai, *Radix cucunorica*’s ecological dominance may explain the limited *Galba* presence. This highlights the importance of controlling *Radix* spp. for disease prevention.

## 4. Discussion

The results revealed that the high-risk areas for *Radix* spp. predominantly lie in south-central Tibet, while those for *F. hepatica* primarily concentrate in the eastern Qinghai Province. These findings align with earlier research [43]. Notably, the non-overlapping high-risk areas of two species suggest high vector efficiency of *Radix* spp. in Qinghai Province, consistent with previous epidemiological studies in the region [10,44,45]. Additionally, the key variables influencing *Radix* spp. distribution were temperature seasonality (Bio4), elevation (Elev), and mean temperature of the driest quarter (Bio9). For *F. hepatica*, the principal determinants were elevation (Elev), human footprint index (Hfp), and precipitation of the driest month (Bio14). Thus, elevation, human activities, and temperature were pivotal environmental factors for both species. Furthermore, the Tibetan goat, the northern Tibetan sheep, Xizang *Bos grunniens*, Qinghai *Bos grunniens*, and Southern Tibetan sheep face higher transmission risks from both *Radix* spp. and *F. hepatica*. This finding validates the direct relationship between grazing practices and *F. hepatica* distribution patterns.

The propagation of the fascioliasis epidemic is primarily dictated by the spatial distribution of freshwater snails, which act as intermediate hosts, favoring their geographical range, environmental circumstances, and grazing patterns [46]. The Lymnaeidae family regroups the principal intermediate hosts for *Fasciola* spp. These hosts favor habitats rich in nutrients and replete with aquatic flora, and are often found crawling on the substratum of water bodies, amidst rocks, or along the shoreline, surrounded by aquatic plants. They are also capable of enduring periods of aridity [47]. *Radix* spp., in particular, are prevalent and possess an average lifespan of approximately a year [48]. *Radix* spp. exhibit a robust adaptability to varying pH levels, flourishing from 5.8 to 9.9, with an optimal pH spectrum between 7.0 and 9.6 [49]. Research has noted the presence of *Radix* spp. even in hypersaline water bodies of northern Tibet, and this species markedly prefers fresh medium salinity environments [50].

Additionally, *Radix* spp. stand out among gastropods for their ability to thrive in extreme cold and tolerate icy waters [51,52]. Thus, *Fasciola* transmission depends on a complex interplay of factors, including climate, topography, elevation, and socioeconomic influences [53]. In summary, this study shows that distribution of *Radix* spp. and *F. hepatica* is markedly influenced by variables such as elevation, temperature, and human interventions. Notably, their distributions overlap significantly (Figure 2). However, *F. hepatica* prevalence does not fully align with its distribution, indicating that not all snails carry the parasite. Tibet has an average altitude above 4000 m and low temperatures. Plateau low temperatures may delay parasites development in snails. Factors like low water temperature can also prevent parasite life cycle from completion in some snail habitats. In contrast, Qinghai has an average altitude of about 3000 m, where high-risk snail distributions in warmer areas better suit *F. hepatica*’s life cycle needs. Previous research typically defined high altitude as below 3000 m [54], but this model extends *F. hepatica*’s suitable habitat to 6000 m. This discrepancy may result from recent Qinghai-Tibetan Plateau warming and increased humidity, which have likely enabled host snails to migrate to higher altitudes. Recent studies have documented high *F. hepatica* infection rates in Qinghai’s Wulan County (with altitudes reaching 5031 m) and Xinhai County (with altitudes reaching 5305 m) [10,55], highlighting the need for comprehensive large-scale studies within Qinghai-Tibet Plateau.

## 5. Conclusions

In the current investigation, the MaxEnt model was employed to predict the potential distribution of *Radix* spp. and *F. hepatica* on the Qinghai-Tibet Plateau and identify their risk areas. The local Centers for Disease Control and Prevention (CDC) should therefore enhance vigilance and implement preventive strategies. For instance, a proactive monitoring strategy should be adopted, triggering interventions when *F. hepatica* infection rate exceeds 0.6 to prevent widespread outbreak. Long-term surveillance could track snail migration patterns under climate change by monitoring the upward expansion of suitable habitats at higher altitudes. Such endeavors are paramount for exploring the Lymnaeidae family, enabling precise targeting of snail populations for sampling and reducing research time and costs. Moreover, the insights from this distributional analysis can guide targeted prevention and control of fascioliasis across the Qinghai-Tibet Plateau, ultimately alleviating livestock industry losses and preserving public health security. Notably, while fascioliasis research is relatively abundant, Lymnaeidae studies on the Qinghai-Tibet Plateau remain scarce, highlighting a significant gap in intermediate host investigation. Future sampling can align with our identified areas to enable comprehensive analysis of Lymnaeidae in this region.

The distribution point data in this study were derived from historical documents and limited field investigations, leading to sparse distribution records. Thus, predictions may contain deviations. In the future, systematic field monitoring should be jointly conducted with disease control and animal husbandry departments to fill data gaps. Additionally, the living environments of *Radix* and *F. hepatica* are complex, with their transmission process affected by various factors (such as water quality, water system distribution, etc.). A closer examination reveals that in assessing the impact on the livestock industry, the lack of species-level precise spatial distribution data for livestock forces reliance on general losses across regions and livestock species, significantly limiting analysis depth and refinement, and maintaining the understanding of industry impacts at a macroscopic and preliminary level. To address these limitations and enhance research reliability and application value, future work should urgently collaborate with professional institutions (such as disease control centers and livestock veterinary stations) to design and implement systematic on-site monitoring plans for filling data gaps. Meanwhile, obtaining high-resolution, species-specific livestock geographic information data will be the key to deepening impact assessments and formulating targeted risk management strategies.

## Figures and Tables

**Figure 1 pathogens-14-00647-f001:**
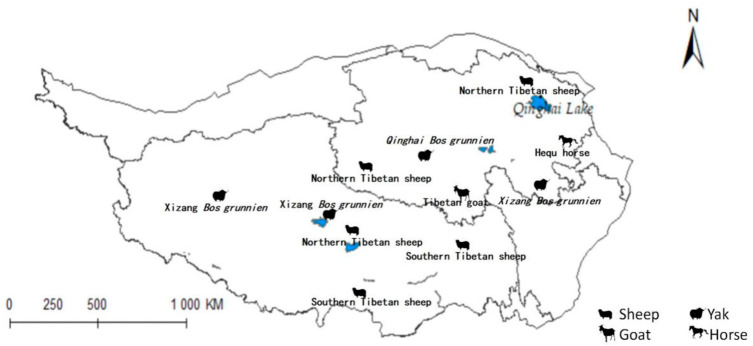
Distribution map of animal husbandry on Qinghai-Tibet Plateau.

**Figure 2 pathogens-14-00647-f002:**
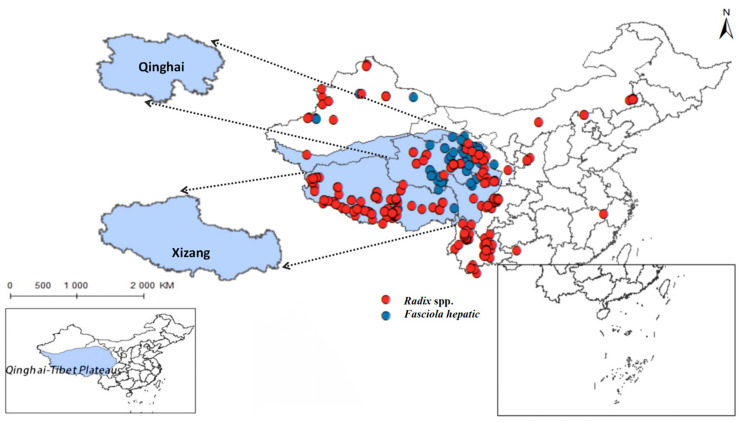
Distribution records of *Radix* spp. and *Fasciola hepatica* on the Qinghai-Tibetan Plateau and neighboring areas.

**Figure 3 pathogens-14-00647-f003:**
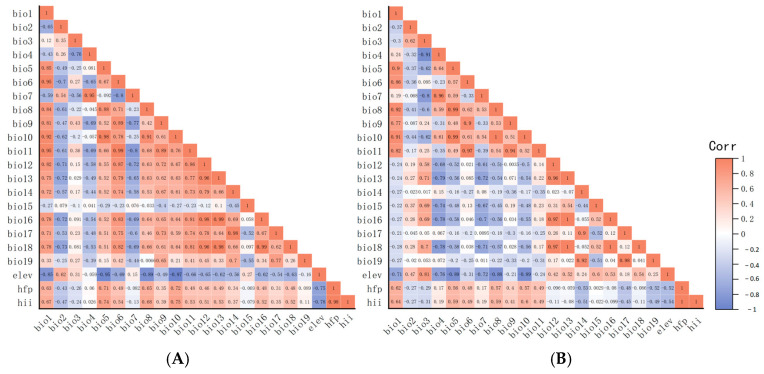
Correlation analyses of environmental factors. (**A**) Correlation analysis of environmental factors of *Radix* spp. (**B**) Correlation analysis of environmental factors of *Fasciola hepatica*. **Note:** Matrices represented by dark orange and dark blue colors indicate environmental variables with absolute correlation coefficients exceeding 0.9. It is recommended to select one of each highly correlated pair for removal.

**Figure 4 pathogens-14-00647-f004:**
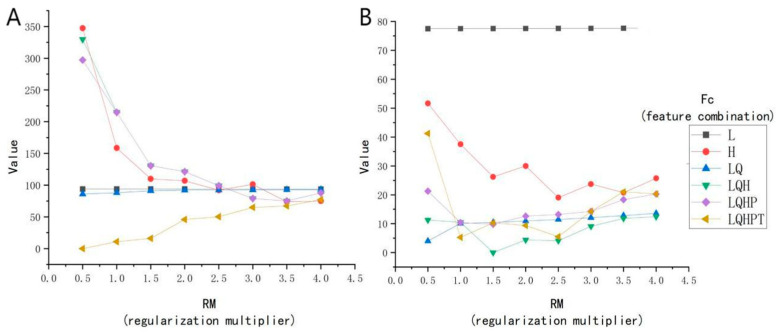
Correlation analyses of environmental factors. (**A**) The *Radix* spp. model of ENMeval packet is used to optimize the results. (**B**) The *Fasciola hepatica* model of ENMeval packet is used to optimize the results. **Note:** L represents linear; Q represents quadratic; H represents hinge; P represents product; and T represents threshold.

**Figure 5 pathogens-14-00647-f005:**
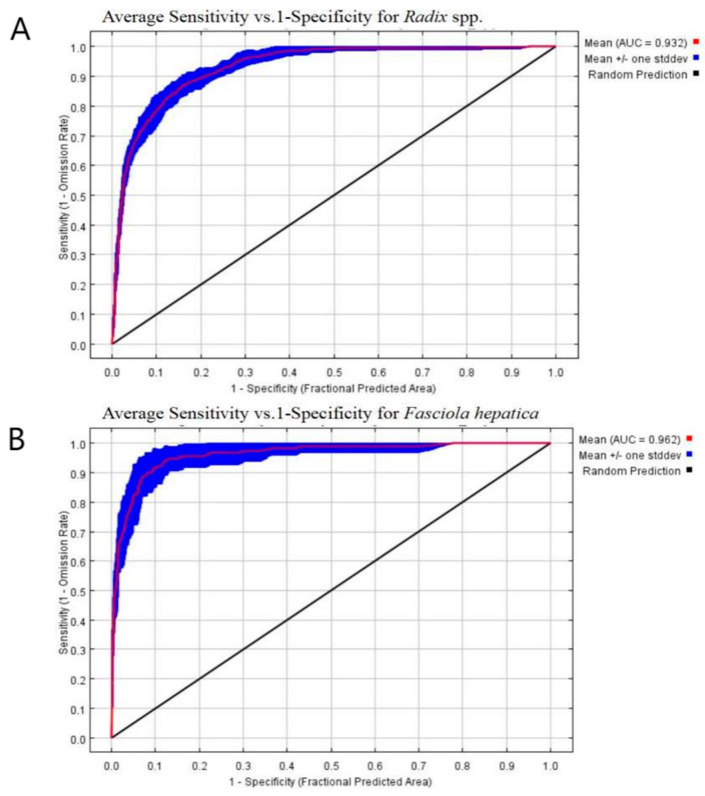
ROC curve analyses. (**A**) The ROC curve analysis of *Radix* spp. (**B**) The ROC curve. analysis of *Fasciola hepatica.*

**Figure 6 pathogens-14-00647-f006:**
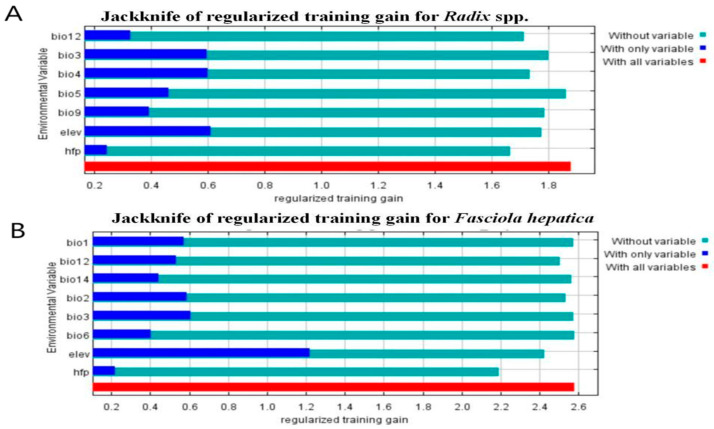
Jackknife test results of the environmental factors affecting the distribution of *Radix* spp. and *Fasciola hepatica*. (**A**) Jackknife test results of the environmental factors affecting the distribution of *Radix* spp. (**B**) Jackknife test results of the environmental factors affecting the distribution of *Fasciola hepatica.*

**Figure 7 pathogens-14-00647-f007:**
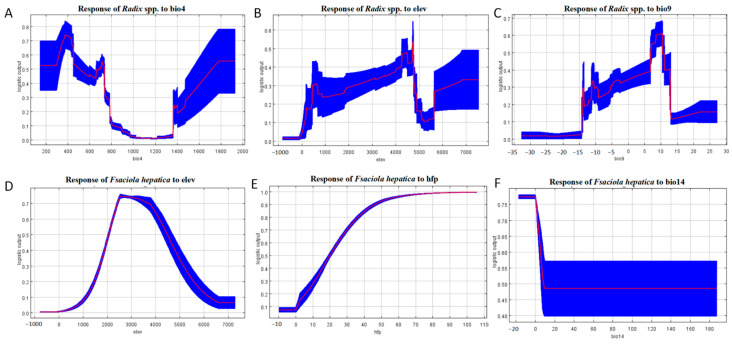
Response curves of the main environmental factors affecting the distribution of *Radix* spp. and *Fasciola hepatica*. (**A**) Response of *Radix* spp. to Bio 4. (**B**) Response of *Radix* spp. to elevation. (**C**) Response of *Radix* spp. to Bio 9. (**D**) Response of *Fasciola hepatica* to elevation. (**E**) Response of *Fasciola hepatica* to human footprint index. (**F**) Response of *Fasciola hepatica* to Bio 14. Note: bio4 represents temperature Seasonality, elev represents elevation, bio9 represents mean temperature of the driest quarter, hfp represents human footprint index, and bio 14 represents precipitation of the driest month. The red line in the figure represents the average value, and the blue block shows the range of this value in the results of ten simulations.

**Figure 8 pathogens-14-00647-f008:**
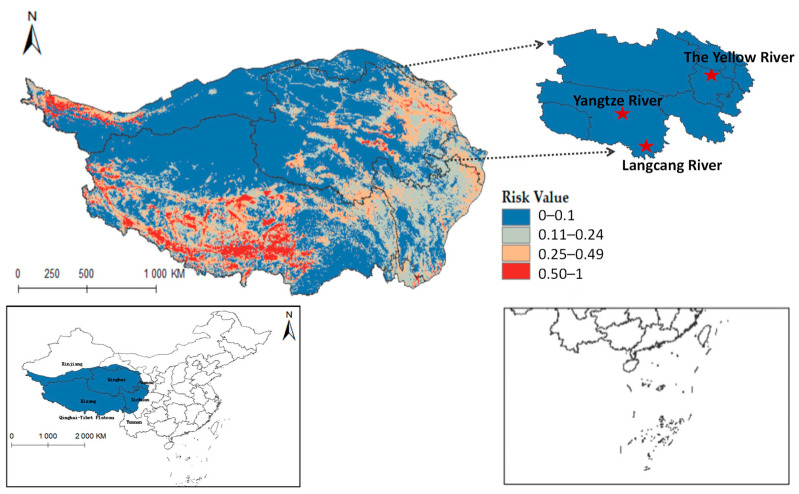
Potential distribution of *Radix* spp. in the Qinghai-Tibet Plateau region.

**Figure 9 pathogens-14-00647-f009:**
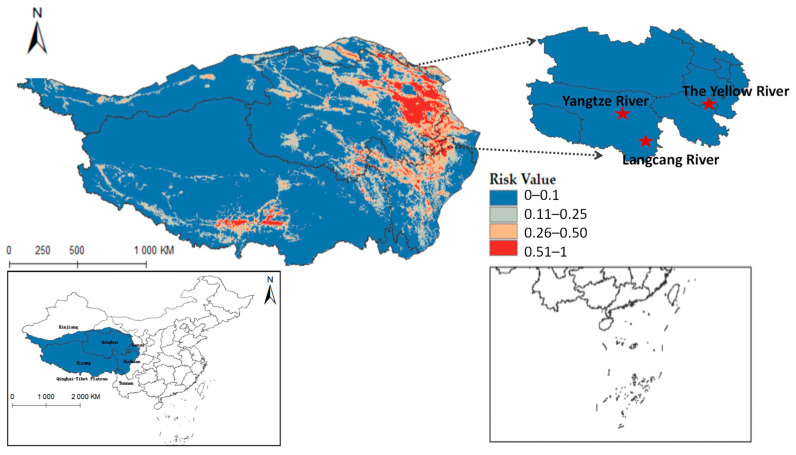
Predicted distribution of *Fasciola hepatica* in Qinghai-Tibet Plateau.

**Figure 10 pathogens-14-00647-f010:**
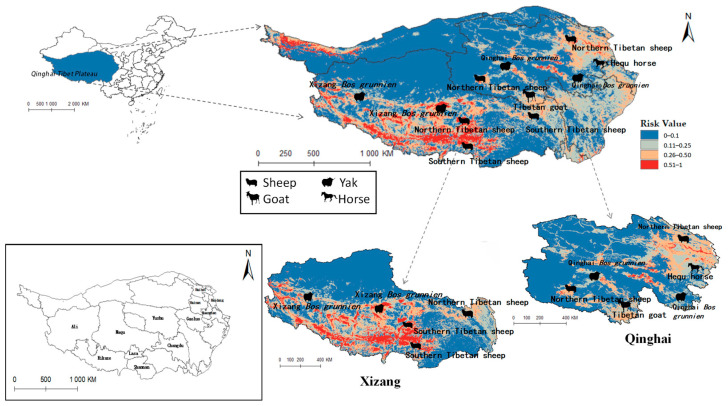
Distribution of animal husbandry in high-risk areas of *Radix* spp.

**Figure 11 pathogens-14-00647-f011:**
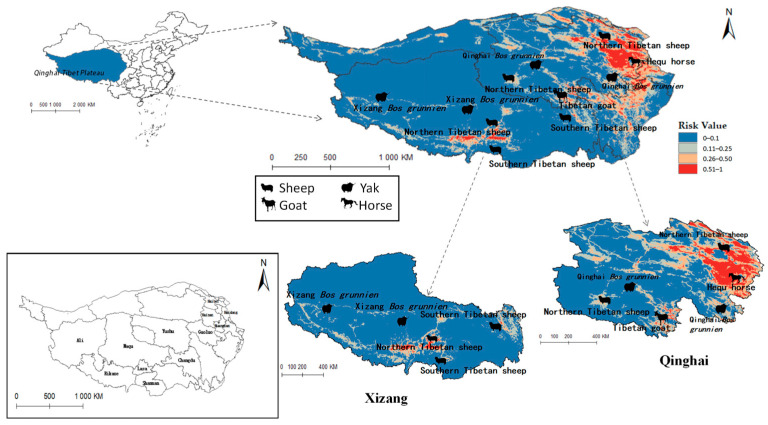
Distribution of animal husbandry in high-risk areas of *Fasciola hepatica*.

**Figure 12 pathogens-14-00647-f012:**
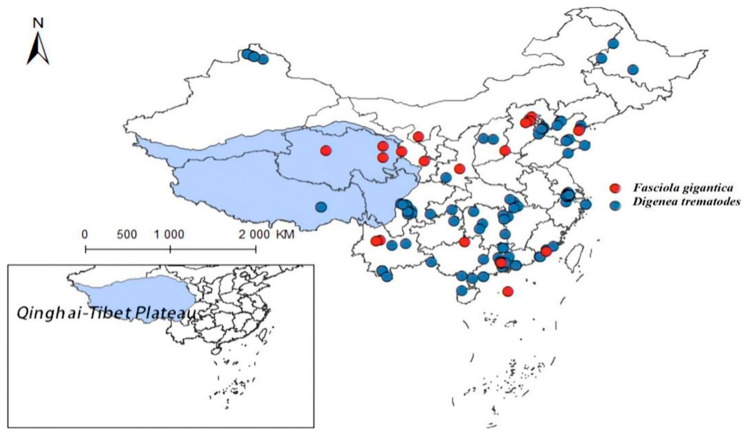
Distribution records of *Fasciola gigantica* and *Digenean trematodes* in China.

**Figure 13 pathogens-14-00647-f013:**
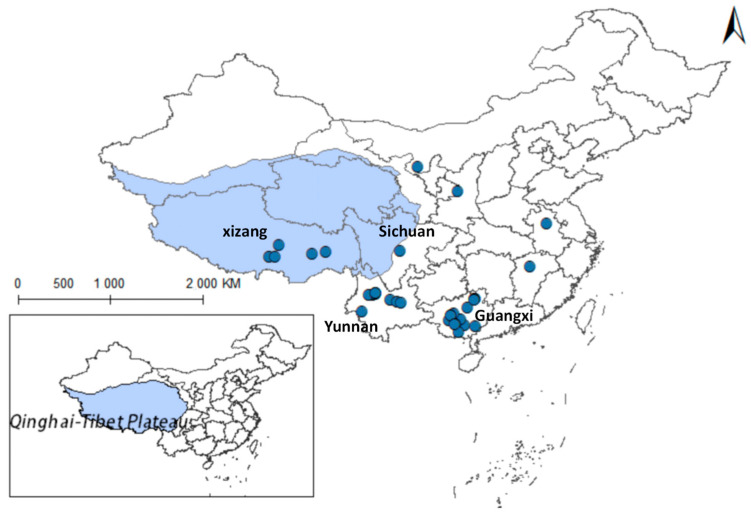
Distribution records of *Galba* spp. in China.

**Table 1 pathogens-14-00647-t001:** Environmental variables and sources.

VARIABLE NAME	DEFINITION	SOURCE
**BIO 1**	Annual mean temperature	WorldClim
**BIO 2**	Mean diurnal range [mean of monthly (max temp-min temp)]	WorldClim
**BIO 3**	Isothermality (BIO 02/BIO 07) × 100	WorldClim
**BIO 4**	Temperature seasonality	WorldClim
**BIO 5**	Maximum temperature of the warmest month	WorldClim
**BIO 6**	Minimum temperature of the coldest month	WorldClim
**BIO 7**	Temperature annual range (BIO 5-BIO 6)	WorldClim
**BIO 8**	Mean temperature of the wettest quarter	WorldClim
**BIO 9**	Mean temperature of the driest quarter	WorldClim
**BIO 10**	Mean temperature of the warmest quarter	WorldClim
**BIO 11**	Mean temperature of the coldest quarter	WorldClim
**BIO 12**	Annual precipitation	WorldClim
**BIO 13**	Precipitation of the wettest month	WorldClim
**BIO 14**	Precipitation of the driest month	WorldClim
**BIO 15**	Precipitation seasonality	WorldClim
**BIO 16**	Precipitation of the wettest quarter	WorldClim
**BIO 17**	Precipitation of the driest quarter	WorldClim
**BIO 18**	Precipitation of the warmest quarter	WorldClim
**BIO 19**	Precipitation of the coldest quarter	WorldClim
**ELEV**	Elevation	WorldClim
**HII**	Human influence index	Last of the Wild Data Version 2 2005
**HFP**	Human footprint index	http://www.ciesin.columbia.edu/wild_areas/ (accessed on 24 March 2024)

**Table 2 pathogens-14-00647-t002:** Distribution records of *Radix* spp. and *Fasciola hepatica* on the Qinghai-Tibetan Plateau and neighboring areas.

LOCATION	SPECIES	SAMPLE SIZE	ELEVATION (AVERAGE VALUE)
**QINGHAI**	*Radix* spp.	35	3000 m
	*Fasciola hepatica*	83	
**XIZANG**	*Radix* spp.	112	4000 m
	*Fasciola hepatica*	6	

**Table 3 pathogens-14-00647-t003:** Contribution and importance ranking of modeling variables of *Radix* spp. (%).

VARIABLE NAME	PERCENT CONTRIBUTION	PERMUTATION IMPORTANCE
**BIO 4**	21.4	32
**ELEV**	16.4	16.9
**BIO 9**	14.7	13.4
**BIO 12**	13.3	22.6
**HFP**	13.2	10.5
**BIO 3**	12.4	3.3

**Table 4 pathogens-14-00647-t004:** Contribution and importance ranking of modeling variables of *Fasciola hepatica* (%).

VARIABLE NAME	PERCENT CONTRIBUTION	PERMUTATION IMPORTANCE
**ELEV**	41.3	47.4
**HFP**	30.5	22.4
**BIO 14**	12.1	1.1
**BIO 2**	11.4	22.1
**BIO 12**	3.8	5.4
**BIO 3**	0.8	0.3

**Table 5 pathogens-14-00647-t005:** Affected animal husbandry situation.

	*Radix*				*Fasciola hepatica*			
Risk Value	High (0.51–1)	Medium (0.26–0.5)	Low (0.11–0.25)	No Risk (0–0.1)	High (0.47–1)	Medium (0.21–0.46)	Low (0.11–0.2)	No Risk (0–0.1)
Northern Tibetan sheep		√			√			
Hequ Horses	√				√			
Tibetan goats	√				√			
Qinghai *Bos grunniens*		√				√		
Xizang *Bos grunniens*	√							√
Southern Tibetan sheep	√				√			

Note: This symbol “√” indicates the presence of a risk.

## Data Availability

Species distribution points used in the construction of the ecological niche model are from existing survey records, specimen records, and relevant databases accumulated over time, such as the Global Biodiversity Information Facility (GBIF; https://www.gbif.org/ accessed on 9 May 2025).

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
