# Peer review of "Projected Distribution and Dispersal Patterns of Potential Distribution Fasciola hepatica and Its Key Intermediate Host Radix spp. in Qinghai-Tibet Plateau, China, Under Plateau Climatic Conditions"

_pathogens, 2025, doi:10.3390/pathogens14070647_

Round 1
Reviewer 1 Report (Previous Reviewer 1)
Comments and Suggestions for Authors
To authors,
The Manuscript ‘Projected distribution and dispersal patterns of prevalent Fasciola hepatica and its key intermediate host Radix spp. in Qinghai-Tibet Plateau, China, under intense climatic conditions / previously titled: Research on the Three-dimensional Coupling Mechanism and Intelligent Assessment of the Transmission Risk of Fasciolosis in the Qinghai-Tibet Plateau (previously titled: Risk assessment of Radix and Fasciola hepatica and their impact on animal husbandry in Qinghai-Tibet Plateau)’ (Manuscript ID: pathogens-3561545_v3; Type: Article; Section: Parasitic Pathogens) presents some problems concerning both content and format:
Methodologically, the manuscript follows a logical sequence. The methodology is adequate, with a sufficient description of the procedures and analytical techniques employed. However, the authors do not provide a detailed explanation of the ‘Species distribution data collection’. I strongly recommend clarifying the use of the classification criteria in the sections ‘Species distribution data collection’ and ‘Livestock impact assessment’, as these are fundamental to the proposed work. If an attempt is made to address the distribution and dispersal patterns of prevalent Fasciola hepatica and the intermediate host Radix spp., only confirmed and valid records of Fasciola hepatica and Radix spp. should be employed.
Results and discussion
Most results are well described and interpreted. Nevertheless, the presentation of data in the section 'Distribution of Radix spp. and F. hepatica, and the criteria used for species definition in the explanation Qinghai–Tibet Plateau and surrounding areas’ is unclear and warrants a more detailed. The projected models for Radix spp. and F. hepatica on the Qinghai–Tibet Plateau, as described in the ‘Prediction of communication risk’ section, are difficult to interpret without detailed information on the presence/absence data and the criteria for species attribution employed in their construction. The considerable morphological plasticity within the molluscan genera analysed—each exhibiting distinct ecological behaviours as well as differing vectorial roles—severely hinders the interpretation of the results. Similarly, the examination of distribution data for Fasciola hepatica and Fasciola gigantica —together with the potential presence of intermediate forms— without clearly defined species‐identification criteria, further compounds the difficulty of assessing the influence of environmental factors that might inform predictive distribution or transmission‐risk models.
The interpretation of the results in the context of the presence of Fasciola hepatica and Fasciola gigantica —together with the potential presence of intermediate forms— cannot be attributed exclusively to a single species, as stated in the manuscript.
This work provides an excellent and indispensable opportunity to analyse of fasciolosis and of freshwater snails acting as intermediate hosts on the Tibetan Plateau. However, a valid identification of the species involved is a prerequisite for examining the epidemiological transmission scenario in high-altitude ecosystems.
Author Response
Please see the attachment.

Reviewer 2 Report (Previous Reviewer 3)
Comments and Suggestions for Authors
Dear authors,
in my view, all the addressed suggestions have now been applied. Nevertheless, please ensure that all abbreviations have been deciphered within the text and pay particular attention to lines 139-140.
Author Response
Please see the attachment.

Reviewer 3 Report (New Reviewer)
Comments and Suggestions for Authors
The study by Xu et al. is devoted to assessing the distribution and dispersal patterns of Fasciola hepatica and its intermediate host Radix spp. in China using machine learning models and approaches. The study appears well-done. The methods are generally well described and make the study sound scientific. Nevertheless, the manuscript needs thorough revision.
- The Introduction is too superfluos, please make it more focused and straightforward. In particular, the last paragraph (lines 121-130) should be removed, as it is already an overview of the results and their significance. In addition, the objectives of the study are not clearly stated.
- Fig. 4. All abbreviations in the figure must be explained in the figure caption.
- Section 3.3. This section of the manuscript is very short, only 4-5 lines, however, Figure 4 is discussed in this section. Thus, the results presented in Figure 4 require more discussion in the text of the section.
- Fig. 7. All abbreviations presented in the figure should be deciphered in the figure caption.
- Section 3.8. I recommend removing all information on 144 species of trematodes, since without detailing it is meaningless, or putting it in the appendices
- Line 416. It is completely unclear what 44 species of Fasciola gigantica are if this is 1 species. Please explain your point of view.
- The manuscript requires such a section as limitations of this study. Please insert such a section at the end of the Discussion section.
Author Response
Please see the attachment.

Reviewer 4 Report (New Reviewer)
Comments and Suggestions for Authors
In the paper by Xu et al. the authors reviewed the various publications and scientific reports that have been published on fasciolosis caused by Fasciola hepatica on the Tibetan plateau in China. Their aim was to determine the factors affecting the transmission of fascioliasis in this region as well as to identify those that have an influence on the presence of the host snail (Radix spp.). The authors then superimposed the distribution of snail populations and that of F. hepatica in relation to the various types of livestock grazing on the Tibetan plateau to analyse the risks to local livestock. This article is very interesting because of the location of these various actors in altitude. Several parts of this text are difficult enough to read because of the many long sentences. On the other hand, the presence of many minor errors in the text of this version requires a major revision.

Author Response
Please see the attachment.

Reviewer 5 Report (New Reviewer)
Comments and Suggestions for Authors
Dear Editor and Authors,
General Overview
This study employs ecological niche modeling (MaxEnt) to predict the potential distribution of Fasciola hepatica and its intermediate host Radix spp. in the Qinghai-Tibet Plateau, China. The research addresses a relevant public and veterinary health issue, applying appropriate methodologies to identify risk areas and assess livestock impacts. The models showed high accuracy (AUC > 0.9) and identified key environmental factors. However, the manuscript presents significant problems in English writing quality, methodological clarity, and results presentation that compromise its comprehensibility and scientific rigor.
Proposed Corrections
Mandatory Revisions
Language Quality and Textual Clarity The manuscript requires comprehensive revision by a native English speaker to address grammatical issues, imprecise scientific terminology, and overly complex sentence structures that impair comprehension.
Methodological Inconsistencies Section 2.2 mentions exclusion of overfitting-prone variables (Bio8, Bio9, Bio18, Bio19), but Bio9 appears as an important variable in results (Table 3). This inconsistency must be clarified. Additionally, the criterion for risk area classification using "Jenks' natural breaks" needs better methodological justification, especially since risk intervals differ between species without clear explanation.
Results Presentation Issues Figures 3A and 3B (correlation analysis) are difficult to interpret due to poor visual quality and inadequate legends. Results in section 3.7 on livestock impacts are primarily descriptive, requiring more robust quantitative analysis.
References and Citations Multiple citation inconsistencies: reference to "Figure 1" on line 227 when discussing livestock distribution that appears later; mentions of "Figure A1" and "Figure A2" that don't correspond to described content; cited supplementary materials (Tables S1, S2) don't match provided descriptions.
Suggested Improvements
Structure and Organization The discussion is excessively long and repetitive, especially between lines 461-495. Condensation focusing on main findings and practical implications is suggested. The introduction could be more concise, eliminating redundancies about fascioliasis importance.
Analytical Depth Authors could better explore implications of distribution differences between Radix spp. and F. hepatica, particularly the high vector efficiency mentioned in discussion. Impact analysis on different livestock species could include quantitative economic risk estimates.
Validation and Limitations The manuscript would benefit from more detailed discussion of model limitations, especially considering relatively small sample sizes for some regions and possible under-representation of data in remote plateau areas.
Final Recommendation
Accept with Major Revisions
The study presents significant scientific merit by addressing an important knowledge gap regarding parasite distribution in an ecologically unique and economically relevant region. The MaxEnt modeling methodology is appropriate and results are technically sound, with high model accuracy (AUC > 0.9). The integration of parasite and host distribution with livestock data represents a valuable contribution to control strategies.
However, identified deficiencies in writing quality, methodological inconsistencies, and results presentation issues require substantial corrections before publication. These revisions are essential to ensure the research achieves its potential scientific and practical impact in veterinary epidemiology and public health.
Comments on the Quality of English LanguageComments on the Quality of English Language
The English language throughout the manuscript requires comprehensive revision to meet publication standards. Several specific issues have been identified:
Terminology and Precision: The title uses vague terminology ("intense climatic conditions") that lacks scientific precision. Terms like "prevalent" in the title context may be misleading - consider "potential distribution" instead. Scientific terminology should be consistently applied throughout.
Sentence Structure and Clarity: Many sentences are overly complex and difficult to follow. For example, lines 113-119 contain convoluted phrasing about high-altitude vulnerability that should be simplified. The abstract contains several grammatically awkward constructions that obscure the key findings.
Word Choice and Flow: Throughout the text, there are issues with article usage, preposition selection, and verb tenses. The writing often lacks natural flow, with abrupt transitions between concepts. For instance, the connection between paragraphs in the introduction needs improvement.
Technical Writing Standards: Scientific writing conventions are inconsistently applied. Some sections use appropriate academic tone while others are overly informal or imprecise. The discussion section particularly suffers from repetitive phrasing and redundant explanations.
Recommendations:
- Engage a native English speaker or professional scientific editor for comprehensive language revision
- Simplify complex sentences while maintaining scientific accuracy
- Ensure consistent use of scientific terminology throughout
- Improve paragraph transitions and overall text flow
- Pay particular attention to the abstract, which should clearly and concisely present the research
While the scientific content is valuable, language barriers currently impede clear communication of the research findings. Professional language editing will significantly enhance the manuscript's impact and accessibility to the international scientific community.
Round 2
Reviewer 4 Report (New Reviewer)
Comments and Suggestions for Authors
Minor revisions.

Author Response
Please see the attachment.

Reviewer 5 Report (New Reviewer)
Comments and Suggestions for Authors
Comments to Authors and Editors - Minor Revisions Required
The authors have addressed most of the major concerns raised in the initial review. However, several specific issues require attention before final acceptance:
**Comment 3 - Figure Quality and Presentation:**
Author Response: "We will replace the pictures here to make them clearer."
**Reviewer Comment:** No changes were observed in Figures 3A and 3B. The figures remain of poor visual quality and difficult to interpret. The correlation matrices still lack adequate resolution and clear legends. Please provide improved, high-resolution figures with proper legends as promised in your response. There is a "note" in the figure, without description. put in this "note", what should be observed in the figures
**Comment 6 - Analytical Depth and Quantitative Analysis:**
Author Response: "However, there are still some deficiencies in this part of our current research."
**Reviewer Comment:** While you acknowledge deficiencies in your livestock risk assessment, these specific deficiencies should be explicitly stated in the Discussion section. Please add a paragraph detailing what these deficiencies are and how they limit your current findings. This transparency will strengthen the manuscript's scientific rigor.
**Comment 7 - Study Limitations (Lines 866-952):**
**Reviewer Comment:** The limitations section should be more focused and comprehensive. Please:
- Concentrate specifically on the limitations of your modeling approach and study design
- Include the deficiencies mentioned in Comment 6 response regarding livestock impact assessment
- Remove any redundant or overly general statements
- Provide specific examples of how data gaps in remote plateau areas might affect your predictions
In addition, do not worry about mentioning in the Discussion that future activities will be carried out, as this weakens the results of your study and comments like this do not enrich your discussion. Address the limitations and future perspectives.
**Additional Recommendation:**
Consider adding a separate **Conclusions** section to better synthesize your main findings and their practical implications for disease control strategies in the Qinghai-Tibet Plateau.
**Overall Assessment:**
The manuscript has improved significantly following revisions. These minor adjustments will enhance clarity and transparency, making the work more valuable to the scientific community. Once these specific issues are addressed, the manuscript will be suitable for publication.
The authors did not respond about the corrections requested. They reported that they made some corrections. But I understand that it still needs to be reviewed by a native English speaker.
Author Response
Please see the attachment.

This manuscript is a resubmission of an earlier submission. The following is a list of the peer review reports and author responses from that submission.
Round 1
Reviewer 1 Report
Comments and Suggestions for Authors
To authors,
The Manuscript ‘Risk assessment of Radix and Fasciola hepatica and their impact on animal husbandry in Qinghai-Tibet Plateau’ (Manuscript ID: pathogens-3561545; Type: Article; Section: Parasitic Pathogens) presents some problems concerning both content and format:
Title
I consider that the title of this manuscript does not fully reflect the work presented; the aspects concerning ‘Radix’, ‘Fasciola hepatica’, and the ‘impact on animal husbandry’ are not comprehensively addressed in the text. The title could either be retained—provided that the methods, results, and discussion sections address these aspects in detail—or revised to better represent the content of the manuscript.
Introduction
Authors should avoid using indirect citations or secondary sources; primary citations, in which the supporting evidence is originally presented, should always be preferred. In this manuscript, the authors aim to predict the potential distribution of Fasciola hepatica, along with its key intermediate host, on the Qinghai–Tibet Plateau. Some errors in taxonomic terminology—both typographical issues and incorrect italicisation—as well as in common names, have been observed, which hinder the interpretation of the text.
Materials and Methods
Methodologically, the manuscript follows a logical sequence. The methodology is adequate, with a sufficient description of the procedures and analytical techniques employed. However, the authors do not provide a detailed explanation of the ‘Species distribution data collection’ – no metadata or meta-information regarding the selection or cleaning protocols of the data classification is supplied. Additionally, the criteria for species definition in the information sources used (morphological, molecular, etc.) are not mentioned. Regarding the ‘Livestock impact assessment’, no meta-information is provided on its influence on animal husbandry in this region. I strongly recommend clarifying the use of the classification criteria in the sections ‘Species distribution data collection’ and ‘Livestock impact assessment’, as these are fundamental to the proposed work
Results and discussion
Most results are well described and interpreted. However, the results of ‘Distribution of Radix spp. and F. hepatica in the Qinghai–Tibet Plateau and surrounding areas’ are not presented in a summary table that allows these findings to be interpreted (e.g. sample size, altitude, location, etc.). The projected models for Radix spp. and F. hepatica on the Qinghai–Tibet Plateau, as described in the ‘Prediction of communication risk’ section, are difficult to interpret without detailed information on the presence/absence data and the criteria for species attribution employed in their construction.
It is essential to incorporate into the analysis the potential records of Fasciola hepatica, F. gigantica and intermediate forms present in the region, as well as data on the molluscs involved in its transmission.
This work represents an excellent and necessary opportunity for the analysis of fasciolosis and of freshwater snails acting as intermediate hosts on the Tibetan Plateau. This study would allow the examination of epidemiological transmission scenarios in high-altitude ecosystems that are strongly influenced by climate change.
Reviewer 2 Report
Comments and Suggestions for Authors
The submitted manuscript entitled “Risk assessment of Radix and Fasciola hepatica and their impact on animal husbandry in Qinghai-Tibet Plateau” presents and interesting findings and is relevant to its field. The methodology is described in detail, making it reproducible. The results are presented with appropriate figures and tables. However, there are some aspects that need to be reviewed and/or modified in order to improve the reading comprehension and information provided. I strongly encourage the authors to address these concerns and resubmit after thorough revision. Below, I detail the comments by sections:
Primary issues:
- Clarity and language: Some sentences are overly complex. Consider simplifying them for better readability. Check for minor grammatical errors and consistency in terminology.
- Scientific language and scientific errors: The manuscript lacks the precision typically expected in scientific literature. Additionally, there are scientific inaccuracies that need correction.
- When using abbreviations, they should be written in full the first time they appear in the main text. You should also review the full text, as several species names are not italicized, and the term spp. is not present in all the provided names of the genus Radix (see title, abstract, Keywords, results, table 4, and discussion sections).
- English proficiency: Awkward sentence structures hinder the manuscript's comprehensibility. The entire article requires editing by a native English speaker for correct use of English.
Abstract:
- Lines 16-23: There is no background, the text is related to material and methods section. Modified it and clarify methodology briefly without excessive detail.
- The novelty or scientific contribution of your study should be indicated in the abstract.
Introduction:
- Lines 36 and 37: correctly specify the taxonomic classification as you made in lines 44 and 45, specify that pond snail is a common name. Also, Lymnaeidae should not be italicized.
- Line 37: Lymnaeidae is a family that includes many species of snails, so you should refer them in plural not in singular. Also, write the number 3 with letters
- Line 38: Lymnaeidae is not the correct name of the genus, modified it.
- Line 42: change “Fascioles” to “caused by species of the genus Fasciola”
- Line 44: Fascioles is not correct, modified it.
- Line 45: the genus is Fasciola not Fascioda. In addition, the species name provided should be written without abbreviations, as this is the first time it is cited in the text.
- Line 47: “fascioliasis hepatica” must be “fascioliasis”
- Line 49: "Fascioliasis hepatica"? What does this sentence refer to, fascioliasis or the species Fasciola hepatica? Please correct it.
- Line 51: Eliminate Lymnaeidae, specify the species
- Lines 51-53: The information provided is disorganized; first, you referred to the family in general, then Fasciola hepatica is cited, and then written 20 species. Rephrase the sentence.
- Line 53: Add a space between spp. and exhibits.
- Line 54: eliminate “our country” and cite the country.
- Line 60: “Fascioles”? correct it.
- Line 62: Add snails after “Lymnaeidae”
- Lines 64-97: Is your work the first to conduct this type of analysis? In this paragraph, you cite examples of predictive model studies for other parasites, but none related to the topic of your study. On the other hand, the information provided in these cited examples could be summarized, and it would be more appropriate to provide previous data related to the topic and location of your research to better justify your work.
- Line 86: The letter P of “Phlebotomous perniciosus” is not italicized.
Materials and Methods:
- If any limitations exist in sample size, data collection, or experimental design, explicitly mention them.
- Line 143: Add and space between variables and Based.
- Line 194: add and space between “Plateau” and “(Figure 1).
- Risk classification into high, medium, low, and no risk levels is mentioned, but it would be helpful to include a brief justification of the thresholds used to define these categories.
Results:
- In section titles and in figure and table captions, write the full species names without abbreviations.
- Ensure that Figures and tables are referenced properly in the text. The number of the figure does not coincident with the number written in text.
- Line 321: The figure 8 caption is combined with the figure. Provide a space to separate the text from the image.
- Lines 333-342: The text mentions locations not listed on the map. It would be helpful for the reader to locate these locations for a better understanding. Same in figure 8.
- Although the results are well-detailed, in some cases the writing is dense and could benefit from a more concise style. It is advisable to rephrase some long sentences to improve the flow of the text.
Discussion:
- The discussion repeats information already presented in the results section, such as the use of the MaxEnt model and AUC values. Instead, it should focus on interpreting the findings and comparing them with previously published data.
- Some sentences are excessively long. I recommended simplifying them to improve understanding.
- There are several grammatical errors that should be corrected, such as “gergraphical” on line 392, “an significant” on line 408 “it indicate” on line 434, among others.
- Although it is mentioned that the results are consistent with previous studies, a deeper analysis of how these findings confirm, contradict or extend the existing literature is lacking.
- The importance of epidemiological surveillance and the role of local CDCs are mentioned, but specific strategies based on the study findings are not detailed.
Reference list:
- The references are not written according to the journal's guidelines. Please correct them according to the instructions for the authors.
- Check that scientific names are in italics and journal names are abbreviated.
Reviewer 3 Report
Comments and Suggestions for Authors
Dear authors,
I have some suggestions for your manuscript.
Minor:
- Please include the relevance or significance of your research in the Abstract section.
- Please accurately check all the figure numbers in the legends.
- Please accurately check all mistakes and typos.
Major:
- Please describe how the mentioned datasets (e.g., annual mean temperature, etc.) were selected (Table1).
- Please highlight the importance of the findings in section 3.5 (figure 6 & Table 2), as this topic appears to have been investigated extensively in prior studies.
- Please explain the importance of using such a large number of different models, given that the ranges of Fasciola and Radix do not significantly overlap according to your data (figure 2). It might be worth considering the other two species of snails, Lymnaeidae and Galba.
- Are all Radix spp. (Figure 10) infected? What does "Province converge with zones of elevated Radix transmission risk" mean?
- Please explain the specific ways in which Radix poses substantial risks to the breeding industries of the Hequ horse and Tibetan goat. The impact of Fasciola on cattle is well understood, but the impact of Radix remains unclear.
- The main comment and question: I do not understand the primary idea of this research. If Fasciola hepatica and Radix spp. do not overlap in their areas, why did the authors create so many combined models for this pond snail and this trematode? The pond snail itself is not dangerous to cattle. Please rewrite this more accurately. Additionally, please combine some pictures into a collage, as it is difficult to perceive them individually.
Round 2
Reviewer 1 Report
Comments and Suggestions for Authors
To authors,
The Manuscript ‘Research on the Three-dimensional Coupling Mechanism and Intelligent Assessment of the Transmission Risk of Fasciolosis in the Qinghai-Tibet Plateau (previously titled: Risk assessment of Radix and Fasciola hepatica and their impact on animal husbandry in Qinghai-Tibet Plateau)’ (Manuscript ID: pathogens-3561545_v2; Type: Article; Section: Parasitic Pathogens) presents some problems concerning both content and format:
Title
I consider that the title of this manuscript does not fully reflect the work presented; the aspects concerning ‘Transmission Risk of Fasciolosis’ are not comprehensively addressed in the text.
Materials and Methods
Methodologically, the manuscript follows a logical sequence. The methodology is adequate, with a sufficient description of the procedures and analytical techniques employed. However, the authors do not provide a detailed explanation of the ‘Species distribution data collection’ – no metadata or meta-information regarding the selection or cleaning protocols of the data classification is supplied. Additionally, the criteria for species definition in the information sources used (morphological, molecular, etc.) are not mentioned. I strongly recommend clarifying the use of the classification criteria in the sections ‘Species distribution data collection’ and ‘Livestock impact assessment’, as these are fundamental to the proposed work
Results and discussion
Most results are well described and interpreted. Nevertheless, the presentation of data in the section 'Distribution of Radix spp. and Fasciola hepatica, and the criteria used for species definition in the explanation Qinghai–Tibet Plateau and surrounding areas’ is unclear and warrants a more detailed. The projected models for Radix spp. and F. hepatica on the Qinghai–Tibet Plateau, as described in the ‘Prediction of communication risk’ section, are difficult to interpret without detailed information on the presence/absence data and the criteria for species attribution employed in their construction. The considerable morphological plasticity within the molluscan genera analysed—each exhibiting distinct ecological behaviours as well as differing vectorial roles—severely hinders the interpretation of the results. Similarly, the examination of distribution data for F. hepatica and F. gigantica —together with the potential presence of intermediate forms— without clearly defined species‐identification criteria, further compounds the difficulty of assessing the influence of environmental factors that might inform predictive distribution or transmission‐risk models.
This work provides an excellent and indispensable opportunity to analyse of fasciolosis and of freshwater snails acting as intermediate hosts on the Tibetan Plateau. However, a valid identification of the species involved is a prerequisite for examining the epidemiological transmission scenario in high-altitude ecosystems.
Reviewer 2 Report
Comments and Suggestions for Authors
The authors have adequately implemented most of the previously suggested changes, and a marked improvement in the quality of the information provided is discernible. However, errors that had already been noted in the previous review remain, especially with regard to the use of taxonomic abbreviations and scientific-style standards.
In particular, it is reiterated that species should be written with the full name the first time they are mentioned in the text as well as when they appear after a period and followed. The use of abbreviations in such cases is not correct and should be avoided. Similarly, names of scientific genera such as Fasciola should always be in italics, according to the standard biological nomenclature. In addition, there are some words that are misspelled.
The following are some specific errors detected:
Line 31: "F. hepatica" should be written as Fasciola hepatica.
Line 33: Correct "estblish" to read "establish".
Line 43: The word "echibited" does not exist in English. It should be corrected by "exhibited".
Lines 68, 71, 73: The name of the genus Fasciola should be in italics.
Line 75: After a period, the names of species should be written without abbreviation.
Line 85: Correct the capital letter in "Being", which should not be capitalized.
Line 88: "R. cucunorrica" must be written in full: Rhipicephalus cucunorrica (check the correct spelling of the specific epithet).
These errors are repeated in different sections of the manuscript, so it is strongly recommended that authors review the text as a whole and correct each instance consistently.
In addition, the manuscript still contains grammatical errors and extensive verbal constructs that make it difficult to understand. A thorough revision of the English language is recommended, paying attention to spelling, punctuation and clarity of sentences. A professional linguistic review or the assistance of a native speaker could be beneficial to ensure the quality of the manuscript.
Reviewer 3 Report
Comments and Suggestions for Authors
Dear authors,
Thank you for revising the manuscript point by point. I am now completely satisfied with the current version of the manuscript and thank you for addressing
all of my suggestions.
I recommend paying attention to typos in the main text.
